# Identifying resources used by young people to overcome mental distress in three Latin American cities: a qualitative study

Mauricio Toyama,[1] Natalia Godoy-Casasbuenas ,[2] Natividad Olivar,[3] Luis Ignacio Brusco,[3] Fernando Carbonetti,[3] Francisco Diez-Canseco,[1] Carlos Gómez-Restrepo ,[2,4] Paul Heritage,[5] Liliana Hidalgo-Padilla,[1] Miguel Uribe,[4] Mariana Steffen,[5] Catherine Fung ,[6] Stefan Priebe[6]

For numbered affiliations see end of article.

**Correspondence to**
Dr Natalia Godoy-Casasbuenas;
natalia.godoy@javeriana.edu.co

## ABSTRACT

**Objective** To explore which resources and activities help young people living in deprived urban environments in Latin America to recover from depression and/or anxiety.

**Design** A multimethod, qualitative study with 18 online focus groups and 12 online structured group conversations embedded into arts workshops.

**Setting** This study was conducted in Bogotá (Colombia), Buenos Aires (Argentina) and Lima (Peru).

**Participants** Adolescents (15–16 years old) and young adults (20–24 years old) with capacity to provide assent/consent and professionals (older than 18 years of age) that had experience of professionally working with young people were willing to share personal experience within a group, and had capacity to provide consent.

**Results** A total of 185 participants took part in this study: 111 participants (36 adolescents, 35 young adults and 40 professionals) attended the 18 focus groups and 74 young people (29 adolescents and 45 young adults) took part in the 12 arts workshops. Eight categories captured the resources and activities that were reported by young people as helpful to overcome mental distress: (1) personal resources, (2) personal development, (3) spirituality and religion, (4) social resources, (5) social media, (6) community resources, (7) activities (subcategorised into artistic, leisure, sports and outdoor activities) and (8) mental health professionals. Personal and social resources as well as artistic activities and sports were the most common resources identified that help adolescents and young adults to overcome depression and anxiety.

**Conclusion** Despite the different contexts of the three cities, young people appear to use similar resources to overcome mental distress. Policies to improve the mental health of young people in deprived urban settings should address the need of community spaces, where young people can play sports, meet and engage in groups, and support community organisations that can enable and facilitate a range of social activities.

## STRENGTHS AND LIMITATIONS OF THIS STUDY

⇒ This study explored which resources young people use to overcome mental distress in three large urban Latin American cities employing Spanish as the same language, which facilitated the analysis across countries and allowed for direct comparisons.

⇒ The study combined different perspectives (adolescents, young adults and professionals) and methods (focus groups and structured group conversations embedded in arts workshops) providing a different approach to engage young people and encourage them to express their views.

⇒ The recruitment of a substantial proportion of participants was done through arts organisations, which may have biased the findings, leading to a particularly strong emphasis on arts activities.

⇒ The study was conducted during the COVID-19 pandemic, which may have influenced the views of young people as to what they find helpful.

⇒ Focus groups and structured group conversations were conducted online, which may have influenced the interaction between the facilitators and the participants and thus the results.

## INTRODUCTION

Adolescence and young adulthood represent a period of growth and development. Young people experience physical changes as well as changes in their cognitions, emotions and social environment.[1,2] At the same time, cultural, socioeconomic and environmental influences, including poverty, low education,[3,4] violence[5] or substance use,[6] make adolescents and young adults particularly vulnerable to developing mental health problems.

According to the WHO, it is estimated that between 10% and 20% of young people experience mental disorders, most commonly depression and anxiety disorders.[7] Depression represents a common of illness and disability in this group, can be associated with substance abuse, aggressiveness and eating disorders and is one of the main risk factors

for suicide. Anxiety can be accompanied by disturbances of sleep, concentration, social and/or occupational functioning,[8] and is the ninth cause of illness and disability in this population.[9] Poor mental health at a young age is associated with further health and social problems such as poorer academic achievement, early school drop-out and substance abuse.[10]

Young people living in major cities across Latin America can be exposed to various risk factors for developing mental disorders, such as poverty, violence and social inequality.[11–14] The percentage of young people with depression and/or anxiety has been reported to range from 17% in Colombia[15] to 26% in Argentina.[16] However, longitudinal studies have found that among young people who suffer from depression and anxiety, between 50% and 60% manage to overcome their distress within 1 year.[17 18]

The ability to overcome mental distress—sometimes referred to as resilience—is linked to personal resources such as motivation and hope and to social resources such as support from peers and access to activities in the surrounding environment.[19 20] In children and young adults, arts activities can be an important social resource.[21] In Latin America, studies about resources to cope with adversity have been carried out in the context of social violence,[2 22 23] poverty,[24] forced displacement[25] and medical conditions,[26] but to the best of our knowledge, no study has explored the resources that help young people in this region to overcome mental distress.

The aim of this study is to explore the participant's perceptions of which resources and activities help young people in large cities in Latin America to recover from mental distress in form of depression and/or anxiety, using both conventional focus groups and arts workshops.

## METHODS
### Study design
This is an exploratory qualitative study using a grounded theory design.[27] The study was conducted within the context of a larger research programme called 'Building resilience and resources to reduce depression and anxiety in young people from urban neighbourhoods in Latin America (OLA)'. The overall aim of the programme is to identify which characteristics, resources and activities help young people living in urban environments in Latin America to prevent or recover from depression and/or anxiety.[28] The programme is a collaboration between Queen Mary University of London in London (UK), Universidad de Buenos Aires in Buenos Aires (Argentina), Pontificia Universidad Javeriana in Bogotá (Colombia) and Universidad Peruana Cayetano Heredia in Lima (Peru). The researchers work in close partnerships with the following arts organisations: Crear Vale La Pena (Argentina), Fundación Batuta and Familia Ayara (Colombia) and Teatro La Plaza (Peru). These organisations specialise in building creative environments, using

classical music, rap music, street art, dance, theatre and visual arts, respectively.

This study used two forms of group activities for data collection: focus groups including adolescents (15/16 years), young adults (20–24 years) and professionals working with young people; and arts workshops with young people (15–24 years). The arts workshops were intended as a different approach to engage young people and elicit their views. Towards the end or following the arts activities, all arts workshops included a structured group conversation to explore resources to overcome anxiety and/or depression. Because of the restrictions due to the pandemic, all groups were conducted online between August and November of 2020.

A total of 18 focus groups were conducted: six in each country and two for each group of participants (adolescents, young adults and professionals). The focus groups lasted between approximately 90 min and 120 min. There were also 12 arts workshops: 4 in Argentina using a combination of creative activities run by the arts organisation 'Crear Vale La Pena', 2 in Colombia using rap and graffiti run by 'Familia Ayara' and 4 with musical activities run by 'Fundación Batuta' and 2 in Peru using theatre and dramaturgical exercises run by 'Teatro La Plaza'. The length of the workshops varied. They usually lasted between 2 hours and 3 hours. Each focus group and arts workshop had a facilitator and a co-facilitator or technical supporter.

### Study setting
The study was conducted in the capital cities of three Latin American countries—Buenos Aires (Argentina), Bogotá (Colombia) and Lima (Peru)—and focused on young people from poorer economic neighbourhoods based on national indicators of income.

### Participants
Three groups of participants were included in the study: adolescents, young adults and professionals with experience in working with young people.

For the focus groups, the inclusion criteria were: 15–16 years or 20–24 years of age, respectively, willingness to share personal experiences within the group, capacity to provide assent/consent, being currently experiencing depression/anxiety defined by having a clinical diagnosis or currently using mental health services and, for the adolescents, consent from a parent/legal guardian.

Professionals were included if they were at least 18 years old, had experience of professionally working with young people, were willing to share personal experience within a group and had capacity to provide consent.

Inclusion criteria for the arts workshops were: 15–24 years of age, current or past participation in activities of the collaborating arts organisations, capacity to provide assent/consent and, for those under 18 years of age, consent from a parent/legal guardian.

## Data collection tools

Basic sociodemographic data and previous experience of anxiety and/or depression were collected from all participants in the focus groups on a brief questionnaire, completed by phone or through videoconference (eg, Zoom).

Topic guides for both focus groups and structured group conversations within the arts workshops addressed the resources used by young people when experiencing mental distress. We introduced the concept of distress using anxiety and depression as examples since they are the most common forms of mental disorders (see online supplemental files 1, 2). The same topic guides were applied in the three countries.

## Procedures

Participants were identified through schools, health services, non-governmental organisations and youth organisations as well as the four participating arts organisations. Potentially interested participants were contacted via phone by the research team to check the inclusion criteria before they were invited to formally participate in the study. The informed consent procedures were conducted through phone or video-conference. Participants were given the options to provide their consent by phone recording, completing the consent form virtually or completing the form on paper and sending a scan or picture to the research team. Adolescents provided their informed assent, and a parent or legal guardian was contacted to receive their permission and consent. The focus groups and structured group conversations were conducted by researchers trained in qualitative data collection and analysis.

## Data analysis

Audio recordings of the focus groups and structured group conversations were transcribed verbatim. We conducted an inductive content analysis using the four stages proposed by Bengtsson.[29] The research team from each country analysed the transcriptions of their local participants. At least two members of the research teams in each country worked together during the analysis. During the first stage, decontextualisation, the research teams familiarised themselves with the data and inductively created codes based on the relevant information they identified. These codes were added to a spreadsheet shared between the teams. During the recontextualisation stage, the codes and categories were discussed in weekly meetings to standardise the codes across the three countries to ensure reliability. After the meetings, each team revisited their transcriptions to either recode their coding to match those identified in other countries, or to code new information they may have missed, based on codes created by the other teams. Furthermore, during the categorisation stage, the codes were discussed and grouped into categories based on similarity of content. These categories were standardised across the research teams. Finally, during the compilation stage, each team conducted a final analysis, summarising the information

extracted for each code in a table. This table was further summarised in a second round to reduce the amount of information. Subsequently, these summaries were combined into a summary encompassing the information from all three countries.

## Patient and public involvement

Adolescents and young adults (aged 15–24 years old) with knowledge and lived experience of mental distress, including depression and anxiety, were included as part of the Lived Experience Advisory Panel (LEAP) in the three cities, where the study was conducted (Buenos Aires, Bogotá and Lima). We received input from the LEAP committee during the design and conduct of the study. The main roles of the LEAP were to provide expert advice to the local research team on all appropriate aspects of the project, including research materials, methods, results and dissemination.

## RESULTS

### Participants' characteristics

A total of 185 participants took part in this study: 111 participants (36 adolescents, 35 young adults and 40 professionals) attended the 18 focus groups, on average approximately six participants per group in each country; and 74 young people (29 adolescents and 45 young adults) took part in the 12 arts workshops. The average number of participants in the workshops varied across the three countries with five in Argentina, six in Colombia and nine in Peru.

The sociodemographic characteristics of all groups are listed for each city in tables 1 and 2.

### Resources identified

The resources and activities that help young people to recover from mental distress were grouped into eight categories: (1) personal resources, (2) personal development, (3) spirituality and religion, (4) social resources, (5) social media, (6) community resources, (7) activities (subcategorised into artistic, leisure, sports and outdoor activities) and (8) mental health professionals. Due to the large amount of data collected, only the most commonly reported resources are presented here.

#### Personal resources

Young people, adolescents and professionals refer to personal resources as tools to overcome mental health problems. These resources can be allocated to four subcategories: relationship with oneself, personal traits, coping strategies and passive strategies.

The first subcategory, relationship with oneself, emphasised the importance of self-awareness and self-esteem. Some participants mentioned that having clear goals builds their confidence and helps them overcome problems.

It is very important to value yourself and accept the problem and know how to handle the situation (…)

**Table 1** Sociodemographic characteristics of young people in focus groups and arts workshops

**Focus groups**

| | Buenos Aires | Bogotá | Lima | Total |
|---|---|---|---|---|
| Age group | | | | |
| Adolescents | 13 (48%) | 9 (50%) | 14 (54%) | 36 (51%) |
| Young adults | 14 (52%) | 9 (50%) | 12 (46%) | 35 (49%) |
| Education level completed | | | | |
| Primary school | 17 (63%) | 9 (50%) | 14 (54%) | 40 (56%) |
| Secondary school | 4 (15%) | 6 (33%) | 9 (35%) | 19 (27%) |
| Higher/technical education | 6 (22%) | – | 3 (12%) | 9 (13%) |
| Other | – | 3 (17%) | – | 3 (4%) |
| Previous experience of anxiety and/or depression | | | | |
| Yes | 17 (63%) | 14 (78%) | 18 (69%) | 49 (69%) |
| No | 10 (37%) | 4 (22%) | 8 (31%) | 22 (31%) |
| Previous experiences with mental health treatment of those who reported previous experience of anxiety and/or depression | | | | |
| Yes | 9 (53%) | 11 (79%) | 7 (39%) | 27 (55%) |
| No | 8 (47%) | 3 (21%) | 11 (61%) | 22 (45%) |
| Arts workshops | | | | |
| Age group | | | | |
| Adolescents | 9 (45%) | 11 (30%) | 9 (53%) | 29 (39%) |
| Young adults | 11 (55%) | 26 (70%) | 8 (47%) | 45 (61%) |

It is also very important to know yourself and know what you like and do not like to know what can help you with the distress. (Female adolescent, Buenos Aires)

The second subcategory, personal traits, refers to what young people consider to be the key in overcoming mental distress. Most commonly, empathy was mentioned as a helpful trait but also resilience and motivation.

The third subcategory were coping strategies, including reflecting about problems, accepting and thinking about a difficult situation, controlling emotions and talking to oneself.

The last subcategory 'passive strategies' involved avoiding or isolating themselves from other people when feeling distressed.

When I feel depressed or sad, I make an abrupt change to the situation, I stop what I am doing, I leave and, for instance, listen to music, draw, or, since generally I do not go out, only sometimes, I prefer to be alone, it is how I feel more peaceful. (Male young adult, Bogotá)

### Personal development

Personal development comprises activities related to academic, occupational and/or vocational development. Young adults mentioned work, job placement and entrepreneurship as a helpful resource to cope with difficult situations and stay active and motivated. School attendance was perceived by professionals as a protective factor for adolescents.

Young people without access to school, or outside educational activities, without access to a recreational space, are negatively affected. Economic factors influence a lot and a lack of resources is closely related to the economic situation. (Psychologist, Buenos Aires)

**Table 2** Sociodemographic characteristics of professionals participating in focus groups

| | Buenos Aires | Bogotá | Lima | Total |
|---|---|---|---|---|
| Psychologists | 4 (29%) | 4 (29%) | 3 (25%) | 11 (28%) |
| Teachers | 3 (21%) | 6 (43%) | 1 (8%) | 10 (25%) |
| Youth workers | 3 (21%) | 2 (14%) | 3 (25%) | 8 (20%) |
| Psychiatrists | 2 (14%) | – | – | 2 (5%) |
| Other (eg, volunteer, researcher and consultant) | 2 (14%) | 2 (14%) | 5 (42%) | 7 (23%) |

## Spirituality and religion

Religion and spirituality were reported as important resources by participants from Bogotá and Lima. For spirituality, some participants from Bogotá stated that meditation as a practice helped them to relax, calm their mind and get rid of negative energy. Some participants from Lima, mostly young women, mentioned exercises related to relaxation, such as breathing exercises, yoga, meditation and mindfulness.

Religion was suggested as a resource by participants from both cities. In Bogotá, religion was seen as a resource in moments of sadness and depression and a way to seek guidance for solving problems. In Lima, some participants talked about praying or reading the bible and talking to God was regarded in a similar manner as meditating.

> I am Christian, but it's not like I go to confession with a priest (laughs). It's more like simply talking with God, I tell Him 'Father, this happened to me'. It's curious because I did not understand very well from a scientific point of view why that happened, but there is research that says if you are talking with someone, even if they don't exist, it is like a direct meditation. That's how I see it. (Female adolescent, Lima)

Some participants from Bogotá pointed out that religion can make people feel judged or not good enough. Some participants from Bogotá and Lima remarked that young people are not as religious as older people.

## Social resources

Social resources focused on receiving support from friends, family and other people, as well as participating in group activities. Participants from all three cities agreed that support from friends is an important source of emotional support for young people. Talking about their problems made them feel better and engaging in activities with friends was a welcome distraction. Professionals highlighted friends and social groups as an important resource, in some cases more important than the family.

> I believe it is fundamental, the support, the emotional support that family or friends can provide. Like someone mentioned, a lot of the time you confide in your friends more than in your family. (Male young adult, Bogotá)

Repeatedly, young people referred to the family as an important source of support, providing helpful advice and making them feel calmer and more relaxed. Family support was seen as a valuable resource to navigate situations of emotional distress. Other social sources of support were partners and teachers.

> In my case, I talk more with my family, my mom, my sisters, my dad, mostly about things that happen at work, I always talk to them. First, because they are closer to me and because I believe they have the experience to give me advice, with other issues, maybe outside of work, I may talk to friends, but almost always I talk to my family. (Male young adult, Lima)

Another helpful resource was the sense of belonging to a group with similar interests and with whom they can share experiences and views. Most frequently reported were sports, artistic and religious groups, but political groups and scouts also came up as helpful resources. These groups helped to forget about problems, openly express feelings, feel understood and supported by peers, and develop skills (ie, artistic skills). Socialising and communicating in these groups could improve the mood in situations of distress.

## Social media

Social media were reported as a valuable resource to distract themselves, maintain contact with friends and acquaintances, meet new people or watch inspiring content. However, some participants also referred to negative aspects of social media such as promoting unrealistic expectations about living standards and physical appearance as well as increase the risk of being cyberbullied.

> It's a bit 50/50, because it depends on the type of people you have on your social network. Why I say this, because in my case, during school, for a long time I was cyber bullied. I had to close my accounts, report it, everything, all bad. And in the same place I found people who supported me and helped me, motivated me and took me through the path of God. I guess it depends on the type of people you have on your social network and how you use it. (Female young adult, Bogotá)

## Community resources

Community resources were suggested only by participants from Bogotá and Buenos Aires. They included parks, cultural centres, botanical gardens, public libraries and activities such as workshops focused on arts, writing, reading, poetry and sports. Young adults and adolescents highlighted the importance of having access to these spaces to meet other people or engage in activities, which help them forget about their problems.

Professionals stated that these resources and spaces helped young people to develop their identity and build bonds, emphasising more often clubs, schools and community spaces.

> When I go to the cultural centre and dance or when I go to the Scouts on Saturday, I focus so much on that [activity] that I forget about my problems and have a great time. (Female adolescent, Buenos Aires)

## Activities
### Arts activities

Arts activities such as playing music, painting, drawing and dance were commonly reported as a resource that allowed participants to meet people with similar interests and express themselves and their emotions freely in

a contained context, particularly in situations of discomfort. Such activities would also help to develop social and problem-solving skills and to gain self-knowledge and confidence. Arts activities were perceived liberating and stress reducing. In addition, professionals stated that arts enable young people to explore and develop their talents.

In addition to the above-mentioned arts activities, writing was suggested as a resource, mostly by women. Some write stories or poems as it allows them to process and express their emotions, while others prefer free writing about their problems or any other topic as a way to relax and see the problems from another point of view. Other less common artistic activities mentioned were theatre, clown, circus, graffiti, rap, sculpture and photography.

> It's not just one thing, not just drawing, not just listening to music, not just writing, it's everything that has to do with art, I feel it liberates me a lot, it gives me peace, calms me down when facing situations so complicated, we have. It is like your own psychologist, to whom you can tell how you feel without thinking of others and that is why I like it so much. (Male young adult, Bogotá)

### *Leisure activities*
Hobbies and enjoyable leisure activities such as listening to music or attending a concert were reported as helping to manage stress, relax, forget about problems and create supportive bonds with others.

> Whenever I felt sad, I listened [to music], I put on my headphones, put some music on that I know I listened to when I felt happy. (Male young adult, Lima)

Other leisure activities that worked as a welcome distraction from problems and distress were watching television, movies, series, music videos and broadcasts of sporting events; manual activities such as cooking, knitting, sewing and recycling; playing with pets; reading; listening to podcasts; attending conferences; solving brain teasers; playing games; and going on trips.

### *Sports activities*
A further resource were sports and physical activities which helped to relax, distract, overcome emotional distress and bond with peers.

> When I do sports, the problems leave, you focus only on what you have to do or what you want to do, to improve and be okay. It clears your head a lot and that helps you overcome any problem. (Male adolescent, Buenos Aires)

### *Outdoor activities*
Some participants mentioned outdoor activities as a resource which helped to relax, particularly at times of sadness and distress. These activities included going out for a walk, either alone or in a group, skateboarding and hanging out or sunbathing with friends in the park.

> When I have been in those moments [of mental distress] I went to the park and seat there, close my eyes and just listen to the sounds, the wind, how the plants move. (Female young adult, Lima)

Other outdoor activities were related to contact with nature such as nature walks, going to parks, camping, going to the beach or the riverside.

> I noticed watching nature helped me a lot, it relaxes me and helps me process things. To see the sky and more natural environments, I noticed it helps me greatly. (Female young adult, Bogotá)

### Mental health professionals
As listed in table 1, more than a third of the young people in the focus groups reported previous experiences with mental health treatment. Their experiences were mostly with psychologists at school, university or youth groups. In Bogotá, some participants also mentioned having participated in support groups and coaching led by psychologists.

The experiences were reported as mostly positive and helpful to solve problems, vent emotions and receive support. Those participants who had never been in contact with a mental health professional, they stated that they would seek contact with a professional only if they had a serious problem, which they could handle on their own.

> If the problem has gotten out of my hands, I would go to a psychologist, for guidance. (Male young adult, Lima)

## DISCUSSION
### Main findings
Adolescents and young adults living in large cities in Latin America report a range of resources that can help them to recover from mental distress. Across the capitals in three Latin American countries, similar resources were identified. They were grouped into eight categories: personal resources, personal development, spirituality and religion, social resources, community resources, social media, activities (artistic, leisure, sports and outdoor activities) and mental health professionals. Several of the categories overlap, but all of them focus on different types of resources. In their combination, they show a rich picture of potential resources that young people from economically disadvantaged districts in large cities in Latin America use and find helpful to cope with mental distress.

The reported resources address personal aspects such as personal development, the expression of emotions, feeling distracted and the acquisition of skills. Most of the resources, however, have a strong social element and refer to support received from family, friends and other groups. Various of the most commonly reported activities

combine the two aspects: they distract and cheer up the young person, and at the same time they involve being in groups and feeling understood and supported by others.

## Strengths and limitations

To the best of our knowledge, this is the first study to explore resources of young people to overcome mental distress in large urban environments across three Latin American countries. Using the same language facilitated the analysis across countries and allowed for direct comparisons. The study captured not only perspectives from three different countries but also from three groups, that is, adolescents, young adults and professionals. Moreover, the material was generated through two different methods. We used focus groups as a more conventional form of qualitative approach and structured group conversations embedded in arts workshops, which is a more innovative method in this context and provides a different approach to engage young people and encourage them to express their views. Combining the different perspectives and methods yielded comprehensive material with many commonalities, suggesting a validity of the findings.

However, this study also had some limitations. First, and most importantly, focus groups and structured group conversations were conducted online. This was inevitable because of the restrictions due to the pandemic. Although this mode of communication posed fewer practical problems than originally expected by the research team and still generated rich material, the online communication may have influenced the interaction between the facilitators and the participants and thus the results. Second, the recruitment of a substantial proportion of the participants through arts organisations may have biased the findings, leading to a particularly strong emphasis on arts activities and related group experiences. Third, we conducted the study with young people from economically disadvantaged districts in three large cities in Latin America. The three cities have some commonalities and differences. The cultural, economic and social context in large cities in other Latin American countries may differ so that not all of the findings of this study may be generalisable to cities in those other countries. Fourth, the study was conducted during the pandemic, which may have influenced the views of young people as to what they would find helpful, perhaps emphasising group activities that were difficult to access during that period. Finally, we assessed the views of young people and professionals about helpful resources, which may not be representative of the experience of all young people in Latin America and have no evidence as to what extent those resources are actually used and are beneficial in reality.

## Comparison with existing literature

Several of the findings can be related to other research and appear largely consistent with what has been found in other studies before. Activities related to academic, occupational or vocational development such as work,

job placement and entrepreneurship were considered as a helpful resource to cope with difficult situations and to help young people stay active and motivated. These findings may be related to the lower rates of anxiety and depressive disorders found in adolescents and young adults with higher educational levels in a study in Chile.[30]

Receiving support from friends and belonging to a group were considered an important social resource.[20] While family was also reported as a helpful resource, this was not as consistently mentioned as one might have expected given the cultural importance of the family in the Latin American context.[31] One can only speculate as to whether this was because of the age of the participants who—as adolescents and young adults—were in a phase of life where they take a distance from their family of origin and have not yet founded a new family themselves.[32]

Arts activities and sports were two resources frequently reported as helpful to improve mental health. Activities such as painting, dance, music, handcrafts and other creative arts were identified as a form of non-verbal expression of feelings that improves confidence and helps to reduce distress. Several studies have identified the benefits of these types of activities for young people's mental health. A systematic review showed that artistic activities are beneficial as they allow participants to communicate through art, express thoughts and feelings more effectively, increase self-esteem and improve the ability to handle anger.[33] Another study showed that teens who take part in weekly extracurricular activities, including art, had less anxiety and depression and were generally more optimistic.[34] Similarly, participants suggested that physical activities such as running and team sports were beneficial to reduce stress and improve energy and mood. This finding is consistent with the results of a systematic review showing sports are associated with an improvement in psychosocial health.[35]

## Implications

The findings may have several implications. For reducing anxiety and depression of an individual or on a group level, not all of the resources listed in this study are likely to be equally available or relevant in a given context. However, the eight categories may provide a framework that can help to consider different options for developing programmes that facilitate and strengthen some of the individual and social resources and activities that may help young people to overcome their distress in lower income urban settings.

The findings show that young people come up with a wide range of ideas and experiences about helpful resources. Thus, when considering approaches to improve the mental health of young people, their voice should be heard and take a central role in the development of policies.[36] When their views are elicited, innovative methods like arts workshops may help to engage young people and generate helpful material. Based on the experiences in this study, conducting such workshops and interviews online appears a feasible and valuable

method to provide a range of findings. Online methods may help to complete similar studies with fewer resources and in shorter periods of time than studies with conventional in-person interviews, thus making them more feasible, particularly in low-resource settings.

Most of the young people in our study reported to have experienced anxiety and/or depression and more than a third had received some type of treatment, although there is no information about the type of treatment. In any case, young people reported largely positive experiences of mental health professionals. Future policies to improve the mental health of young people in large cities should address the need for accessible and effective mental health services. However, this cannot be the sole focus as the provision of mental health services requires substantial funding and using professionals can come with the implicit message that the affected young person cannot cope on their own, which in turn may undermine confidence rather than strengthen it. On a population level, promoting personal and interpersonal skills at school, support from peers and groups, and access to various activities appear more important to reduce the overall level of mental distress in young people.

Finally, considering that this study was carried out in urban environments, it is important to stress the value of community spaces where young people are able to play sports, meet, engage in enjoyable group activities and distract themselves such as parks, public libraries or theatres. In addition to the accessibility of appropriate space, there should also be community organisations enabling and facilitating different activities, especially since some sports and arts activities require equipment and guidance. Future research may explore to what extent community organisations should be set up and supported to attract, engage and help particularly vulnerable and mentally distressed young people.

## CONCLUSION

Anxiety and depression are frequent among young people. This study shows that in a challenging context like large cities in Latin America, there is a wide range of resources, similar across the three cities, that young people can use and report as helpful. Future research and policies may focus on how these resources can be made widely available, be strengthened and be flexibly used by young people if and when required.

**Author affiliations**
[1]CRONICAS Center of Excellence in Chronic Diseases, Universidad Peruana Cayetano Heredia, Lima, Peru
[2]Department of Clinical Epidemiology and Biostatistics, Pontificia Universidad Javeriana, Bogotá, Colombia
[3]Department of Psychiatry and Mental Health, School of Medicine, Universidad de Buenos Aires, Buenos Aires, Argentina
[4]Department of Psychiatry and Mental Health, Pontificia Universidad Javeriana, Bogotá, Colombia
[5]School of English and Drama, Queen Mary University of London, London, UK
[6]Unit for Social and Community Psychiatry, Queen Mary University of London, London, UK

**Contributors** SP, LIB, FD-C, CG-R, CF, MU and PH devised the project, the main conceptual ideas and design of the study. MT, NG-C, NO, FC, LH-P, CG-R, MU, MS and FD-C participated in data collection during the artistic workshops and focus groups. MT, NG-C, LH-P, NO, CF and FC participated in the transcription and data analysis of the structured group conversations and focus groups. MT, NG-C and NO wrote the manuscript with input from all authors. SP is responsible for the overall content of the paper participating as the guarantor.

**Funding** This work was supported by the Medical Research Council (grant number: MR/S03580X/1).

**Competing interests** None declared.

**Patient and public involvement** Patients and/or the public were involved in the design, or conduct, or reporting, or dissemination plans of this research. Refer to the Methods section for further details.

**Patient consent for publication** Not applicable.

**Ethics approval** This study involves human participants and was approved by the Faculty of Medicine, Research and Ethics Committee of the Pontificia Universidad Javeriana, Bogota on 20 March 2020 (FM-CIE-0241-20), the institutional ethics committee on research of the Universidad Peruana Cayetano Heredia, Lima, on 31 March 2020 (Constancia E021-03-20), institutional ethics committee of the Universidad de Buenos Aires on 10 February 2020 and ethics of research committee of Queen Mary on 16 November 2020 (QMERC2020/02). Participants gave informed consent to participate in the study before taking part.

**Provenance and peer review** Not commissioned; externally peer reviewed.

**Data availability statement** Data are available upon reasonable request. The consent forms did not specify that the data would be deposited in a public repository, and for this reason we are unable to deposit the data. The de-identified participant dataset will be made available from the corresponding author (NG-C, natalia.godoy@javeriana.edu.co) upon reasonable request and subject to a data sharing agreement.

**ORCID iDs**
Natalia Godoy-Casasbuenas http://orcid.org/0000-0002-1262-6436
Carlos Gómez-Restrepo http://orcid.org/0000-0002-9013-5384
Catherine Fung http://orcid.org/0000-0002-3220-6930

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
