## [Reviewer comments · BMJ Open]

ARTICLE DETAILS

TITLE (PROVISIONAL)	Identifying resources used by young people to overcome mental distress in three Latin American cities: A qualitative study
AUTHORS	Godoy-Casasbuenas, Natalia; Toyama, Mauricio; Olivar, Natividad; Brusco, Luis Ignacio; Carbonetti, Fernando; Diez-Canseco, Francisco; Gómez-Restrepo, Carlos; Heritage, Paul; Hidalgo-Padilla, Liliana; Uribe, Miguel; Steffen, Mariana; Fung, Catherine; Priebe, Stefan

VERSION 1 – REVIEW

REVIEWER	Shepherd, Andrew University of Manchester
REVIEW RETURNED	04-Feb-2022

GENERAL COMMENTS	Dear Colleagues, Many thanks for the opportunity to review this interesting manuscript addressing the strategies used by adolescents and young adults in the face of mental distress, within three Latin American cities. The authors present an extensive qualitative inquiry – drawing on focus groups and individual interviews with young people and practitioners. The findings from this are of potential value in identifying support strategies in the face of growing difficulties and increasing challenges in mental health care provision. The findings are resonant with wider literature in this field and I found myself agreeing with the authors as to the importance of this work. However, I had some concerns regarding the representation and description of the methodology that would prevent me from recommending publication at this time. In terms of overarching comments I felt there was a lack of depth of consideration for the methods applied and the context in which this investigation took place. I would have also welcomed a more thorough engagement with academic literature in relation to overlapping fields – for example the extensive literature on personal recovery which overlaps significantly with the data presented here. Specific comments are summarised below: 1. Introduction p4 line 56: “irritability, dizziness, chest pain, insomnia, fatigue...” – why were these signs picked to describe experience of anxiety disorder? These are non-specific phenomena?2. Introduction and throughout: - There is an emphasis on “arts activities” that is potentially appropriate however the choice of setting – in arts classes etc – does seem to suggest this is likely informed by the population chosen for investigation here?3. Aim – the aim of the investigation is to explore recovery experiences in “...mental distress in form of depression and / or anxiety...” – this is not reflected in the recruitment strategy which
--

	makes no reference to anxiety / depression as experiences sought amongst participants? 4. Qualitative study design – limited or no information is given about the nature of the methodology employed and theoretical underpinnings drawn upon? 5. Focus groups / interviews – These are different interview / data collection methods and generally seen as developing different forms of information / data. Combining them in this way is confusing – was there any distinction in how these methods were handled? 6. Cross-national research – Researchers are drawn from different countries but little information is presented as to how this trans-national perspective is built upon and explored. In fact the methodology presented seems to attempt to bracket out any such exploration? 7. Findings – You quote the “most commonly reported”, I would expect some more justification of this decision as the recruitment strategy reported is not statistical in nature. 8. Presented quotes – How were illustrative quotations selected? With thanks
--	---

REVIEWER	Barrera, Alvaro University of Oxford, Psychiatry
REVIEW RETURNED	09-Feb-2022

GENERAL COMMENTS	Thank you. This is an excellent article. Just one questions. How did the authors ascertain that the participants were from actual poor backgrounds? Was it a geographical criteria? Could this be discussed? Some minor editing: Lines 46 to 48 need to be reviewed. Line 54: is the comma correct?
--

VERSION 1 – AUTHOR RESPONSE

Comment 7:

R1-1: Introduction p4 line 56: “irritability, dizziness, chest pain, insomnia, fatigue...” – why were these signs picked to describe experience of anxiety disorder? These are non-specific phenomena?

Answer:

We included more specific symptoms and a reference Page 3 Lines 11-12

Comment 8:

R1-2: Introduction and throughout: - There is an emphasis on “arts activities” that is potentially appropriate however the choice of setting – in arts classes etc – does seem to suggest this is likely informed by the population chosen for investigation here?

Answer:

We appreciate the comment. However, the choice of setting responds to a logistic reason. We aimed to include participants who were already involved in arts activities. Due to the ongoing COVID-19 pandemic, our only access to this population was through our partners, which are renowned arts organizations in each country, to help us with the recruitment for this activity.

Comment 9:

R1-3: Aim – the aim of the investigation is to explore recovery experiences in “...mental distress in form of depression and / or anxiety...” – this is not reflected in the recruitment strategy which makes no reference to anxiety / depression as experiences sought amongst participants?

Answer: We have adjusted the phrasing of the aim to specify we are focusing on the perceptions of the participants regarding the resources and activities that are helpful to recover from mental distress. Therefore, having experienced anxiety or depression was not an inclusion criterion. Page 3 Line 30

Comment 10:

R1-4: Qualitative study design – limited or no information is given about the nature of the methodology employed and theoretical underpinnings drawn upon?

Answer:

We have added more information regarding the study design. Page 3 Line 35

Comment 11:

R1-5: Focus groups / interviews – These are different interview / data collection methods and generally seen as developing different forms of information / data. Combining them in this way is confusing – was there any distinction in how these methods were handled?

Answer:

We appreciate the comment. However, there seems to be a confusion about the data collection methods. Both focus groups and structured conversations embedded in the arts workshops were group activities. They are both quite similar data collection methods, the only difference is the structured conversations were part of a workshop, and the activities in the workshop aimed to ease the participants into the conversation to facilitate data collection. We understand how this may generate confusion; therefore, we have added the specification of “structured group conversations” throughout the manuscript.

Comment 12:

R1-6: Cross-national research – Researchers are drawn from different countries but little information is presented as to how this trans-national perspective is built upon and explored. In fact the methodology presented seems to attempt to bracket out any such exploration?

Answer:

We appreciate the comment. The aim of the research was to identify resources that do not depend on the possibly very specific context of one city, but have some level of generalizability, at least across the large capitals in Latin America which share a number of characteristics. We therefore conducted the study in deprived areas of three Latin American countries and focused in our analysis on commonalities across various participants/groups both within and across countries. We collected and analyzed the information using the same methods across the three countries. The study was not designed to identify specific differences as this would have required a different methodology. In our case, some resources may not have been as prominently mentioned in one country as in another, but we refrained from interpreting such differences as differences in resources used, since we cannot make that generalization. Not applicable

Comment 13:

R1-7: Findings – You quote the “most commonly reported”, I would expect some more justification of this decision as the recruitment strategy reported is not statistical in nature.

Answer:

We appreciate the comment. This is an exploratory qualitative study that aimed to identify the resources and activities that help young people to recover from mental distress. Therefore, the results present the most common or agreed resources mentioned by the participants, which is a standard procedure when presenting qualitative results. There is no allusion that these results are representative of all young people. We have added an explicit statement about this in the limitations

section. Page 14 Lines 5-6

Comment 14:

R1-8: Presented quotes – How were illustrative quotations selected?

Answer: We selected the quotes based on how descriptive they were while also being succinct to avoid excessive text.

Comment 15:

R2-1: How did the authors ascertain that the participants were from actual poor backgrounds? Was it a geographical criteria? Could this be discussed?

Answer:

We have added this specification. We used national indicators of economic income to identify poorer neighborhoods in which we focused our recruitment efforts. Page 4 Line 29

Comment 16:

R2-2: Lines 46 to 48 need to be reviewed.

Answer:

We appreciate the observation. However, since the reviewer did not specify the page, we are not able to identify which lines he is referring to.

Comment 17:

R2-3: Line 54: is the comma correct?

Answer:

We appreciate the observation. However, since the reviewer did not specify the page, we are not able to identify which lines he is referring to.

Finally, we welcome further comments, if any.

VERSION 2 – REVIEW

REVIEWER	Shepherd, Andrew University of Manchester
REVIEW RETURNED	04-May-2022

GENERAL COMMENTS	Dear Colleagues, Many thanks for the further opportunity to review this paper addressing the recovery experiences of individuals accessing community organisations with respect to their experiences of mental distress. As previously, in my last review, I retained some overarching concerns regarding the depth and transparency of the analysis employed, as well as some clarity with respect to the aims of the study, recruitment strategy, and significance of certain themes in the findings: 1. Focus on experience of anxiety and depression: - You state in your response to my previous review that you have altered the aims / objectives of this study to clarify that you were not focusing on experiences of anxiety and depression per se but instead on a broader construct of mental distress. By my reading this focus remains however – with reference to anxiety and depression being retained where you say it was removed? Also, the focus of the introduction highlights the significance of symptoms of anxiety?
---

	Please clarify as to how this aligns with your recruitment strategy as before? 2. Qualitative study design – You have added a statement that you followed a grounded theory design and then provide reference to your thematic analysis strategy. I still think that there is a lack of detail here with respect to theoretical underpinning and how issues of reflexivity were addressed in your analysis. 3. Presentation of most common themes – While I agree this is a common practice I still think this needs to be remarked upon as it implies some degree of emphasising responses that are said more frequently than others, which risks underplaying significant themes. As a qualitative study the recruitment strategy is not statistical in its nature – and appropriately so – but I do feel that some comment here is warranted noting that all themes were considered and that no priority is presented on the basis of spurious pseudo-statistical arguments. With all best wishes
--	--

VERSION 2 – AUTHOR RESPONSE

Comment # 1:

Ed: Focus on experience of anxiety and depression: - You state in your response to my previous review that you have altered the aims / objectives of this study to clarify that you were not focusing on experiences of anxiety and depression per se but instead on a broader construct of mental distress. By my reading this focus remains however – with reference to anxiety and depression being retained where you say it was removed? Also, the focus of the introduction highlights the significance of symptoms of anxiety? Please clarify as to how this aligns with your recruitment strategy as before?

Response:

As we tried to explain in our previous response letter, we re-phrased the aim of the study to clarify that we are focusing on the perceptions of the participants regarding the resources and activities that are helpful to recovering from mental distress in form of depression and anxiety, whilst we did not study the experiences of anxiety and depression as such. We used anxiety and depression as the most common indicators of mental distress (as anxiety and depression are also the most common mental disorders) and believe our aim to assess perceptions of how to overcome mental distress and our recruitment strategy are consistent.

We added in the manuscript that anxiety and depression were used as inclusion criteria for the recruitment of young adults and adolescents for the focus groups.

Page and paragraph: Page 5: Line 1-2 and lines 14-16

Comment #2:

Qualitative study design – You have added a statement that you followed a grounded theory design and then provide reference to your thematic analysis strategy. I still think that there is a lack of detail here with respect to theoretical underpinning and how issues of reflexivity were addressed in your analysis.

Response: We appreciate the comment and agree that a richer theoretical background might have been helpful. However, no more sophisticated theoretical underpinning was actually used in the study.

Therefore, we are reluctant to retrospectively claim more theoretical guidance than we had really used in the study.

Comment #3:

Ed: Presentation of most common themes – While I agree this is a common practice I still think this needs to be remarked upon as it implies some degree of emphasising responses that are said more frequently than others, which risks underplaying significant themes. As a qualitative study the recruitment strategy is not statistical in its nature – and appropriately so – but I do feel that some comment here is warranted noting that all themes were considered and that no priority is presented on the basis of spurious pseudo-statistical arguments.

Response: We appreciate the concern that mere frequencies may have dominated the analysis and can reassure you that this was not the case. Still, as explained in the paper, in this study we searched for commonalities across participants, and in such search frequency is one aspect to consider, even if not the only or even main one. However, a theme that is mentioned by all or most participants would be considered differently than a theme that was raised by only one or two participants. In our experience with similar analyses, this consideration of frequencies is inevitable and appropriate.

We hope that the paper will be acceptable for publication in your journal.